# Bone Differentiation Ability of CD146-Positive Stem Cells from Human Exfoliated Deciduous Teeth

**DOI:** 10.3390/ijms24044048

**Published:** 2023-02-17

**Authors:** Ryo Kunimatsu, Kodai Rikitake, Yuki Yoshimi, Nurul Aisyah Rizky Putranti, Yoko Hayashi, Kotaro Tanimoto

**Affiliations:** 1Department of Orthodontics and Craniofacial Developmental Biology, Graduate School of Biomedical and Health Sciences, Hiroshima University, 1-2-3 Kasumi, Minami-ku, Hiroshima 734-8553, Japan; 2Analysis Center of Life Science, Natural Science Center for Basic Research and Development, Hiroshima University, 1-2-3 Kasumi, Minami-ku, Hiroshima 734-8553, Japan

**Keywords:** stem cells from human exfoliated deciduous teeth, bone regeneration, CD146

## Abstract

Regenerative therapy for tissues by mesenchymal stem cell (MSCs) transplantation has received much attention. The cluster of differentiation (CD)146 marker, a surface-antigen of stem cells, is crucial for angiogenic and osseous differentiation abilities. Bone regeneration is accelerated by the transplantation of CD146-positive deciduous dental pulp-derived mesenchymal stem cells contained in stem cells from human exfoliated deciduous teeth (SHED) into a living donor. However, the role of CD146 in SHED remains unclear. This study aimed to compare the effects of CD146 on cell proliferative and substrate metabolic abilities in a population of SHED. SHED was isolated from deciduous teeth, and flow cytometry was used to analyze the expression of MSCs markers. Cell sorting was performed to recover the CD146-positive cell population (CD146+) and CD146-negative cell population (CD146-). CD146 + SHED without cell sorting and CD146-SHED were examined and compared among three groups. To investigate the effect of CD146 on cell proliferation ability, an analysis of cell proliferation ability was performed using BrdU assay and MTS assay. The bone differentiation ability was evaluated using an alkaline phosphatase (ALP) stain after inducing bone differentiation, and the quality of ALP protein expressed was examined. We also performed Alizarin red staining and evaluated the calcified deposits. The gene expression of ALP, bone morphogenetic protein-2 (BMP-2), and osteocalcin (OCN) was analyzed using a real-time polymerase chain reaction. There was no significant difference in cell proliferation among the three groups. The expression of ALP stain, Alizarin red stain, ALP, BMP-2, and OCN was the highest in the CD146+ group. CD146 + SHED had higher osteogenic differentiation potential compared with SHED and CD146-SHED. CD146 contained in SHED may be a valuable population of cells for bone regeneration therapy.

## 1. Introduction

Regenerative medicine is a medical technology that uses stem cells to regenerate tissues that have become dysfunctional; it was developed as a new therapeutic technology to replace organ and bone transplantation [1,2,3,4]. Mesenchymal stem cells (MSCs) were first identified as colony-forming cells with the ability to differentiate into osteoblasts, adipocytes, and chondrocytes within bone marrow organs [5]. MSCs are present in skeletal muscles, adipocytes, placentae, dental pulp, and periodontal ligament, and play roles in preparing for and maintaining homeostasis during the restoration of compromised tissues [6,7]. In addition, since MSCs can be collected from tissues and grown in standardized culture conditions, they are used as a transplanted cell preparation for autoimplantation in the medical field; cell preparation for skin and cartilage regeneration has already been marketed [8,9]. Notably, techniques for the isolation and culture of MSCs from oral tissues have been developed in dentistry, and research and development for the practical application of regenerative treatment of dental pulp and periodontal tissues have been promoted [10,11,12].

Since the beginning of 2000, a search for MSC sources in intraoral tissues has been undertaken. Studies have demonstrated and identified the presence of tissue stemness in stem cells from human exfoliated deciduous teeth (SHED) and dental pulp stem cells (DPSCs) [13,14]. SHED and DPSCs have a higher proliferative potential than bone marrow-derived MSCs (BMSCs), and their potential to differentiate into osteoblasts, adipocytes, chondrocytes, and neural cells has been demonstrated [15,16,17,18]. SHED and DPSCs are involved in the formation of dentin and pulp complexes [11,15] and bone [19,20]. Previous studies reported that the successful isolation rates of SHED and DPSCs were approximately 83% and 70%, respectively [21], and SHED and DPSCs had a bone regeneration capacity equivalent to that of BMSCs when SHED, DPSCs, and BMSCs were seeded in the polylactic acid membrane and transplanted into the parietal bone defect model of immunodeficient mice [22]. Kunimatsu et al. found that SHED had a higher cellular proliferative capacity than DPSCs and BMSCs, and SHED showed significantly higher expression of fibroblast growth factor (bFGF) and bone morphogenetic protein-2 (BMP-2) genes compared with hDPSCs and hBMSCs [23]. Moreover, SHED reportedly have a higher capacity for adipogenic and osteogenic differentiation compared to DPSCs [24]. In addition, SHED have been suggested to be distinctly different in nature from DPSC in recent review articles; they also have a higher osteogenic differentiation potential compared to DPSCs [25]. Kichenbrand et al. reported the following benefits of SHED: 1. Hard tissues cover the pulp; thus, less DNA injury occurs in the pulp, 2. It is easy to harvest MSCs, and the procedure is painless and non-invasive, 3. Several tissue samples can be collected, and 4. Since permanent teeth replace deciduous teeth, no ethical concerns exist [26]. In addition, SHED possess advantages for tissue regeneration, which allows quick in vitro expansion before implantation of the tissue, compared to their DPSCs [27]. Thus, SHED is becoming an attractive source of cells and a potential candidate for tissue regeneration. However, the cell population involved in the bone regeneration mechanism of SHED has not yet been elucidated.

Recently, the surface antigens of MSCs have garnered attention, as many surface antigens of MSCs serve as coreceptors for growth factors and provide valuable benefits for the regeneration of various tissues, such as promoting angiogenesis and osteogenesis [28]. MSCs isolated from tissues are a heterogeneous cell population expressing various surface antigens. The effect of surface antigens on cell properties can be investigated using cells with the surface antigen of interest, isolated by cell sorting [29]. Cluster of differentiation (CD)146, a surface antigen, is expressed on MSCs and the plasma membranes of vascular endothelial cells and vascular pericytes, where it functions as a key cellular adhesion molecule in angiogenesis [30,31]. CD146-positive cells isolated by cell sorting from heterogeneous populations of bone marrow- and umbilical cord-derived MSCs have higher bone regenerative potential than CD146-negative cells, and CD146 is related to bone regenerative potential [30,31,32,33]. Therefore, we focused on investigating the expression of CD146 in SHED. In a bone defect immunodeficient mouse model, osteogenesis was promoted by the transplantation of CD146-positive SHED cells [34]. However, a detailed examination of the in vitro effect of CD146 on bone regenerative capacity has not yet been reported in SHED. Accordingly, the present study aimed to compare the cell proliferative and osteogenic potentials of CD146-positive cells, CD146-negative cells, and SHED heterogeneous cells.

## 2. Results

### 2.1. Surface-Antigen Analyses of Isolated SHED

The representative results of one of five donors are shown in Figure 1a. Cells expressing CD146 accounted for 70.9 ± 4.3% of the heterogeneous population of SHED cells isolated from the deciduous pulp. MSC-positive markers, CD90 and CD73, were expressed in all donors, and CD105 was expressed in 99.56 ± 0.37% of donors. CD14, CD19, CD34a, and CD45 (MSC-negative markers) were expressed in 0.08 ± 0.02%, 0.27%, 0.05 ± 0.01%, and 2.93 ± 0.54% of donors, respectively (Figure 1a).

### 2.2. Cellular Proliferative Capacity of SHED, CD146 + SHED, and CD146-SHED

#### 2.2.1. PDT

PDT was 44.06 h, 46.88 h, and 47.15 h in SHED, CD146-SHED, and CD146 + SHED, respectively. However, there were no significant differences among the three groups (Figure 1b).

#### 2.2.2. BrdU Proliferation Assay

Although the proliferative capacity of SHED was slightly higher than that of CD146 + SHED and CD146-SHED 2 h and 24 h after BrdU administration, there were no significant differences among the three groups (Figure 1c,d).

### 2.3. Osteogenic Differentiation-Related Gene Expression Analyses in SHED, CD146 + SHED, and CD146-SHED

Before the induction of osteogenic differentiation, the gene expression of ALP, BMP-2, and OCN did not differ significantly among SHED, CD146 + SHED, and CD146-SHED (Figure 2a). However, CD146 + SHED had significantly higher gene expression of ALP, BMP-2, and OCN on days 21 and 28 after osteodifferentiation induction than SHED and CD146-SHED (Figure 2b,c). In addition, SHED showed significantly higher gene expression of ALP, BMP-2, and OCN than CD146-SHED (Figure 2b,c).

### 2.4. Osteogenic Differentiation Potential of SHED, CD146 + SHED, and CD146-SHED

ALP staining performed on day 21 after inductive osteogenic differentiation revealed a decreased order in the intensity of dark staining in CD146 + SHED, SHED, and CD146-SHED (Figure 3a). In addition, the mean ALP protein levels measured on day 21 after the induction of osteogenic differentiation were 14.28 ± 0.745 μg/mL, 10.27 ± 0.636 μg/mL, and 6.96 ± 0.573 μg/mL in CD146 + SHED, SHED, and CD146-SHED, respectively. CD146 + SHED showed significantly higher ALP protein levels than SHED and CD146-SHED (Figure 3b); SHED showed significantly higher ALP protein levels than CD146-SHED (Figure 3b).

### 2.5. Comparative Analysis of Calcium Deposition among SHED, CD146 + SHED, CD146-SHED

Alizarin red staining was dense in CD146 + SHED, SHED, and CD146-SHED on day 28 after the induction of osteogenic differentiation (Figure 3c), with a decreased staining intensity observed in CD146 + SHED, SHED, and CD146-SHED, in that order. In addition, the absorbance assay showed that CD146 + SHED had significantly higher levels than SHED and CD146-SHED, indicating an increased calcified deposit in CD146 + SHED (Figure 3d).

## 3. Discussion

MSCs are isolated from various tissues, such as the bone marrow, adipose tissue, umbilical cord, and dental pulp, and implanted in defective tissues to promote tissue restoration and regeneration. Since deciduous teeth are shed spontaneously with the permanent tooth replacing them, harvesting SHED is less invasive than harvesting BMSCs, adipose-derived MSCs, and umbilical cord-derived MSCs. Since SHED and BMSCs have a similar bone regeneration capacity, the present study investigated SHED and BMSCs as sources of cells for transplantation into bone defects.

In tissues where abundant blood vessels are present, such as the pulp, there is a microenvironment around the blood vessels called the MSC niche [35,36], which comprises MSCs, hematopoietic stem cells, mesenchymal progenitors, fibroblasts, and pericytes (vascular pericytes) [37]. CD146 is expressed in MSCs and the plasma membrane of vascular endothelial cells and pericytes, and is associated with cell-to-cell or cell-to-extracellular matrix [38,39]. The surface antigen CD146 expressed on MSCs is a receptor for growth factors, such as Netrin-1, Wnt-1, and vascular endothelial growth factor (VEGF)-C [40], and it reportedly acts as a coreceptor for VEGF receptor-2 (VEGFR2) and platelet-derived growth factor receptor beta, thereby contributing to angiogenesis and vascular maintenance [41,42]. Since MSCs and pericytes are present in MSC niches and express some similar surface antigens, including CD146, pericytes are considered the origin of MSCs, and CD146 + MSCs may be very close to pericytes [43,44,45].

In the present study, the surface-antigen analyses of SHED-like cells revealed that more than 99.5% of the heterogeneous cells expressed CD105, CD73, and CD90. However, less than 3% of these cells expressed CD14, CD19, CD34, and CD45. SHED isolated from the pulp of deciduous teeth met the requirements for defining MSCs established by the International Society for Cellular Therapy [46]. CD105, CD73, and CD90 are expressed in these cells under standard culture conditions; however, CD45, CD34, CD14, CD11b, CD79a, CD19, and HLA-DR are not. Although CD105, CD73, and CD90 are MSC-positive markers, they are expressed in MSCs, hematopoietic stem cells, and blood cells [47]. However, MSC-negative markers, i.e., CD14, CD19, CD34, and CD45, are specific markers for hematopoietic stem cells and hemocytes [47]. In the present study, cells isolated from the deciduous pulp were CD105-, CD73-, CD90-, CD14-, CD19-, CD34-, and CD45-negative, and they were very likely to be SHED. Differentiation of SHED (which were isolated in a similar procedure) into osteoblasts, adipocytes, and chondrocytes in vitro systems was confirmed by Miura et al. [13] and Nakajima et al. [23]. In this study, the analysis of CD146 expression rates by flow cytometry showed that CD146 was expressed in approximately 70.9% of SHED heterogeneous populations. In contrast, CD146 was expressed in 48.39–66.3% of SHED heterogeneous populations [48,49]. In this study, adequate numbers of CD146 + SHED were isolated from SHED by cell sorting, which might contribute to slightly higher outcomes than previously reported (70.9% vs. 48.39–66.3%). These findings demonstrate the potential of CD146 + SHED as a valuable source of cells for future clinical applications.

The properties of SHED, CD146 + SHED, and CD146-SHED were investigated in cellular studies. Based on the BrdU cell proliferation assay and PDT analysis, there were no significant differences in proliferative capacity among the three groups, although SHED had a slightly higher proliferative capacity. CD146 + MSCs exhibited significantly higher cell proliferative capacity than CD146-MSCs in human endometrium-derived MSCs and periodontal ligament-derived MSCs [50,51]. However, some papers have demonstrated that CD146-MSCs have higher proliferative potential than CD146 + MSCs [42]. Paduano et al. reported that the proliferative capacity of CD146- MSCs and CD146+ MSCs, cells varied in each study [52]. In the present study, there were no significant differences in the proliferative ability among CD146-SHED, CD146 + SHED, and SHED. Factors accounting for the difference between the reports may include the type of MSC, performance of cellular conditioning or sorting devices used, and effect of the procedure. In this study, cells were passaged after cell sorting, cultured to confluence, and seeded in 96-well and 12-well plates for cell growth testing. However, CD146 + SHED and CD146-SHED grew more slowly immediately after cell sorting than SHED without cell sorting. The longer operating times after releasing cells from the dish, and several centrifugations during the cell sorting procedure, might have minor adverse effects on CD146 + SHED and CD146-SHED. Further investigations with different conditions, such as decreasing the number of passages after cell sorting, are warranted.

Comparative analysis of bone differentiation potential showed that CD146 + SHED had a higher bone differentiation potential than SHED and CD146-SHED based on ALP staining, quantitative ALP analysis, and alizarin red staining of SHED cultured in an osteogenic differentiation-inducing medium. During the osteogenic differentiation-inducing process, the progenitor cells are differentiated into osteoblasts and express ALP from the early- to mid-stage, followed by the expression of bone-specific OCN and, ultimately, the production of hydroxyapatite and collagen I with a crystallographic architecture present in the bone matrix in vivo [53,54,55].

MSC-differentiated osteoblasts are known to secrete VEGF [56]. As an autocrine factor, VEGF secreted by osteoblasts differentiated from SHED may have promoted the osteogenesis of SHED. VEGFR2 is expressed in CD146-SHED [57] and promotes the expression of bone morphogenetic protein-2 (BMP-2) and Runx2, and CD146 acts as a coreceptor for VEGFR2 in CD146 + SHED. In addition, DPSC bone differentiation is promoted by VEGF in vitro [58]. Moreover, the genetic analysis on days 21 and 28 after the initiation of osteogenic differentiation showed that CD146 + SHED had significantly higher gene expression of ALP, OCN, and BMP-2 than SHED and CD146-SHED, indicating that CD146 + SHED has a higher osteogenic differentiation potential. However, to confirm the relationship between VEGF and CD146 + SHED osteogenic differentiation potential, it is necessary to further examine the signaling pathway; expression analysis of genes (such as VEGF, VEGFR2, Runx2, and Osterix) in osteodifferentiated SHED, CD146 + SHED, and CD146-SHED; and osteogenic differentiation potential by treatment with VEGF and anti-VEGF neutralizing antibodies. Previously, it has been reported that hMSC-CD146(+) cells exhibited greater chemotactic attraction in a transwell migration assay, and when injected intravenously into immune-deficient mice following closed femoral fracture, exhibited wider tissue distribution and significantly increased migration ability as demonstrated by bioluminescence imaging [30]. Therefore, CD146 defines a subpopulation of hMSCs capable of bone formation and in vivo trans-endothelial migration and thus represents a population of hMSCs suitable for use in clinical protocols of bone tissue regeneration [30]. Moreover, newly formed bone matrix with embedded osteocytes of donor origin was reportedly observed upon transplantation of CD146(+) human umbilical cord perivascular cells-Gelfoam-alginate 3D complexes in severe combined immunodeficiency (SCID) mice [32]. In addition, a high expression of CD146 in MSCs from bone marrow reportedly correlates with their robust osteogenic differentiation potential [33]. This study suggested CD146 + SHED had superior bone regeneration potential compared to SHED and CD146-SHED in vitro. Based on the present and previous findings on CD146 + BMSCs [30,32,33,39,41,42,52,59], CD146 + SHED may have the following properties: “They have very close properties to pericytes,” and “the binding of VEGF and VEGFR2 further enhances the pathway to promote the expression of bFGF, BMP-2, Runx2, and Osterix as CD146 acts as a coreceptor of VEGFR2.” These factors may have promoted angiogenesis and bone regeneration. A previous study on BMSCs with single-cell sorting of BMSC populations revealed osteo-, adipo-, and chondroid differentiation of as many as 100 cells in clonal culture, and only 50% of these cells differentiated into these three lineages. Furthermore, 80% of BMSCs differentiated into the three lineages expressing CD146, and 40% of the cells differentiated into only one or two lineages expressing CD146 [30]. Therefore, even if a population of CD146-positive cells is isolated, there may be a heterogeneous presence of cells within that population, leading to different differentiation potentials [30]. Thus, it is conceivable that CD146 + SHED and CD146-SHED cell populations isolated from heterogeneous SHED cell populations in this study also differ in function and nature, and may be heterogeneous cell populations. At present, it is difficult to investigate the functional heterogeneity of MSC populations in-depth, and much remains to be elucidated [60]. However, single-cell sorting and clonal culture should also be performed in SHED, CD146 + SHED, and CD146-SHED, and the extent of osteodifferentiation and the ability to differentiate into the three lineages in CD146 + SHED and CD146-SHED populations used in this study should be examined in detail in future studies.

## 4. Materials and Methods

### 4.1. SHED Isolation and Culture

Pulp tissues were collected from deciduous teeth extracted from five healthy patients (Average 9 years 8 months, ± 2 years 4.8 months) who provided informed consent at the Department of Orthodontics, Hiroshima University Hospital. SHED were isolated and cultured using a previously described procedure [21,22,23,34], with reference to the methods of Miura et al. [13] and Gronthos et al. [14]. The following text elaborates on the isolation of SHED from the pulp. A mixture of α-MEM (Sigma-Aldrich, St. Louis, MO, USA), 4 mg/mL collagenase (Thermo Fisher Scientific, Waltham, MA, USA), and 3 mg/mL dispase (Godo Shusei, Tokyo) was prepared. The pulp tissue was immersed in the solution and minced with a scalpel. Adequate dental pulp tissue slices were transferred to a 10-mL tube and then incubated at 37° C under 5% CO_2_ for 20 min with shaking. Cell aggregates were eliminated using a 70-μm cell strainer (CORNING, Corning, NY, USA), and the filtered solution was diluted with α-MEM and centrifuged at 1500 rpm for 5 min. The supernatant was aspirated and added to 20% fetal bovine serum (FBS) (Daiichi Kagaku, Tokyo), 0.24 μL/mL kanamycin (Meiji Seika Pharma Co., Ltd., Tokyo), 0.5 μL/mL penicillin (Meiji Seika Pharma), and 1 μL/mL. After being suspended in α-MEM containing mL amphotericin (MP Biomedicals), they were seeded in a cell culture petri dish (CORNING) with a diameter of 35 mm, cultivated at 37°C and 5% CO_2_, and the cells were detached from the petri dish using PBS containing 0.25% trypsin (Nacalai Tesque, Kyoto) and 1 mM EDTA (Wako Pure Chemical Industries, Osaka) when confluent and passaged. After the first passage (P1), the cells were cultured in α-MEM containing 10% FBS (Daiichi Kagaku, Tokyo) and the abovementioned antibiotics at 37°C under 5% CO_2_. This study was conducted in accordance with the Regulations for Epidemiological Studies of Hiroshima University Hospital (approval no. E-20-2). Cells were independently isolated from the deciduous teeth obtained from the five patients, and the cells were cultured separately.

### 4.2. Fluorescence-Activated Cell Sorting

MSCs were isolated from the pulp of deciduous teeth, and surface-antigen analysis was performed to confirm the presence of CD146. Each SHED collected from 5 patients was cultured and passaged to P3. Flow cytometry was subsequently performed using one of the 10 cm dishes in which the cells of each donor were cultured (five in total), and the targeted surface antigens were analyzed. The targeted surface antigens were CD146 and MSC-positive (CD73, CD90, and CD105) and MSC-negative (CD14, CD19, CD34, and CD45) markers, as defined by the International Society for Cellular Therapy. The cultured cells were detached using phosphate-buffered saline (PBS) containing 0.25% trypsin and 1 mM ethylenediaminetetraacetic acid. The cell suspension was centrifuged at 1800 rpm for 5 min. After aspirating the supernatant, the cells were washed with PBS containing 2% fetal bovine serum (FBS). Two PBS solutions (2% FBS) containing 1 × 10^6^ cells were prepared for each antibody detection.

One of the prepared solutions was supplemented with 30 μL of PE mouse anti-human CD146, PE mouse anti-human CD90, FITC mouse anti-human CD105, PE-CTM 7 mouse anti-human CD14, APC-H7 mouse anti-human CD19 (BD Pharmingen, San Jose, CA, USA), Brilliant Violet™ 421 mouse anti-human CD73, FITC mouse anti-human CD34, and APC-H7 mouse anti-human CD45 (Becton Dickinson, San Jose, CA, USA). In addition, 5 μL of the corresponding kappa isotype control was added to the other prepared solution and incubated at 4 °C, protected from light, for 20 min. Thereafter, the mixture was washed twice with PBS containing 2% FBS and 3 μL of 7-amino-actinomycin (7-AAD; BD Pharmingen) was added. FLOWJO software (Tomy Digital Biology Co., Tokyo, Japan) was used to analyze the surface antigens. The cells were sorted based on the surface antigen analysis to separate CD146-positive (CD146 + SHED) and negative (CD146-SHED) cells using FACS Aria II Cell Sorter (BD Biosciences, San Jose, CA, USA). The separated cells were cultured in α- minimum essential medium (α-MEM) at 37 °C in 5% CO_2_.

The remaining cells were not used for flow cytometry and were used for subsequent experiments as an unsorted SHED group. Following the analysis of these surface antigens by flow cytometry, the cells were merely sorted for CD146, only into CD146-positive and CD146-negative SHEDs. Therefore, surface antigens other than CD146 were only analyzed and not sorted. In subsequent experiments, three groups of CD146-positive and CD146-negative SHEDs isolated by cell sorting and unsorted SHEDs without flow cytometry were used. The resulting SHED cells were cultured separately. Thereafter, we examined the proliferative and osteogenic differentiation activities of each of the five samples of SHED of an individual.

### 4.3. Properties of SHED, CD146 + SHED, and CD146-SHED

#### 4.3.1. Cellular Proliferative Capacity of SHED, CD146 + SHED, and CD146-SHED

The following variables were examined to compare and examine the cellular proliferative abilities of SHED, CD146 + SHED, and CD146-SHED.

##### Population Doubling Time

SHED isolated from deciduous pulp, and CD146 + SHED and CD146-SHED isolated by cell sorting, were cultured in corresponding media, and passage-4 cells were seeded in 24-well plates (CORNING Inc., Corning, NY, USA; 1.0 × 10^4^ cells/well) and cultured in 5% CO_2_ at 37 °C. Dead cells were stained with 0.4% trypan blue (MP Biomedicals, Santa Ana, CA, USA), and the number of live cells was counted daily from day 1 to day 10 of culture using a hemocytometer. Subsequently, a cell growth curve was generated, and the logarithmic growth phase was defined as days 2–6. Population doubling time (PDT) was calculated using the following equation [23].
PDT = (t − t0) log2/logN − logN0
where t0 indicates the time taken for the cell count and the number of cells at N, N0:t, t0.

###### Bromodeoxyuridine Cell Proliferation Assay

Cell growth ELISA and Cell Proliferation ELISA Bromodeoxyuridine (BrdU) kits (Roche Diagnostics, Basel, Switzerland) were used. SHED, CD146 + SHED, and CD146-SHED were seeded in 96-well plates (CORNING; 3 × 10^3^ cells/well) and cultured at 37°C in 5% CO_2_. After 48 h of growth, the cells were incubated with BrdU for 24 h at 37 °C in 5% CO_2_. The absorbance was measured at the wavelength of 375 nm using a microplate reader (MultiskanTM FC; Thermo Fisher Scientific, Waltham, MA, USA).

#### 4.3.2. Induction of Osteogenic Differentiation of SHED, CD146 + SHED, and CD146-SHED

SHED isolated from primary dental pulp, and CD146 + SHED and CD146-SHED isolated by cell sorting, were cultured in corresponding media, and passage-4 cells were seeded in 24-well plates (CORNING) coated with type I collagen (Nippon Ham, Osaka, Japan; 1.0 × 10^4^ cells/well) and cultured at 37 °C in a 5% CO_2_ incubator. After the cells reached 80% confluence, 100 nM dexamethasone (Sigma-Aldrich), 0.2 mM ascorbate-phosphate (Sigma-Aldrich), and 10 mM β-glycerolphosphate (Tokyo Kasei Kogyo, Tokyo, Japan) were added to the cell culture media (D-MEM; Sigma-Aldrich). The osteogenic differentiation-inducing media were changed every 2 days.

### 4.4. Quantitative Real-Time Polymerase Chain Reaction Analysis

SHED, CD146 + SHED, and CD146-SHED were cultured at 37°C under 5% CO_2_. After the cells reached 80% confluence, induction of differentiation was initiated with the osteodifferentiation induction medium described above. The cells were harvested before induction, and at 21 and 28 days after induction. The mRNA expression levels of alkaline phosphatase (ALP), osteocalcin (OCN, a bone transcription factor), and bone morphogenetic protein-2 (BMP-2) were determined using quantitative real-time polymerase chain reaction (RT-PCR) analysis with QuantiTect SYBR Green PCR master mix (Qiagen, Valencia, CA, USA) with a LightCycler^®^ 480 II instrument (Roche Diagnostics). Total RNA was extracted from cells using an RNeasy Mini kit (Qiagen) and quantified using a NanoDrop One/Onec spectrophotometer (Thermo Fisher Scientific, Inc., Waltham, MA, USA). RNA purity was also assessed using this instrument based on the OD 260/OD 280 ratio; only samples with an A260/A280 ratio of 1.5–2.0 were used for further analysis. Subsequently, 1 μg of purified total RNA was reverse-transcribed to cDNA using a ReverTra Ace first-strand cDNA synthesis kit (Toyobo, Osaka, Japan). RT-PCR was performed using Thunderbird SYBR qPCR mix (Toyobo) with specific primer sets (Table 1).

### 4.5. ALP Staining and Determination of ALP Activity

The cells were fixed on days 3, 7, 14, 21, and 28 after incubation with an osteogenic differentiation medium, and ALP staining was performed using the following methods. After fixation with 4% paraformaldehyde PBS (Fujifilm Wako Pure Chemical, Osaka, Japan) for 10 min, the cells were incubated with PBS containing 0.05% Tween-20 (Roche Diagnostics) (washing buffer). After removing the washing buffer, the cells were incubated with ALP staining solution (Fujifilm Wako Pure Chemical Industries, Ltd.) at room temperature under light-resistant conditions for 10 min.

For the quantitative testing of ALP, the cells were harvested on days 3, 7, 14, 21, and 28 using the pNPP Phosphatase Assay Kit (AnaSpec, Fremont, CA, USA) after incubation with an osteogenic differentiation induction medium. The harvested cells were homogenized using a Sonic Vibra Cell (Sonic & Materials, Newtown, CT, USA), and the supernatant was collected and used as the sample. The sample was mixed with 50 μL of the pNPP substrate solution in a 96-well plate (CORNING), and the absorbance was determined at a wavelength of 405 nm using a microplate reader MultiskanTM FC (Thermo Fisher Scientific).

### 4.6. Calcium Deposition Analyses (Alizarin Red Staining)

After the cells were cultured with an osteogenic differentiation-inducing medium for 28 days, they were washed with PBS and 10 mM Tris-HCl (pH 7.5) with 0.9% NaCl. The cells were fixed with 4% paraformaldehyde and stained with 1% Alizarin Red S (Kshida Chemical, Osaka, Japan). The stained tissue sections were photographed and observed using a BZ-X810 microscope (Keyence, Osaka, Japan). In addition, for quantification analyses, the cells were incubated with a mixed solution of 10% acetic acid and 20% methanol at room temperature for 15 min after staining to elute the stained dye. The eluate was added to a 96-well plate (CORNING), and the absorbance was determined at a wavelength of 405 nm using a microplate reader Multiskan™FC (Thermo Fisher Scientific).

### 4.7. Statistical Analysis

All data are presented as mean ± standard deviation. The Kruskal–Wallis test, a nonparametric test, was performed to analyze significant differences between the groups using the software BellCurve^®^ for Excel (SSRI; Tokyo, Japan). *p* < 0.05 and < 0.01 were considered statistically significant.

## 5. Conclusions

In conclusion, the results indicated that CD146 + SHED has superior bone regeneration ability compared with SHED and CD146-SHED. Moreover, CD146 affects the bone regeneration ability of SHED, and CD146 + SHED might be helpful for bone regeneration treatment. Further studies are needed to elucidate the detailed mechanism of bone regeneration in CD146 + SHED for the clinical application of SHED.

## Figures and Tables

**Figure 1 ijms-24-04048-f001:**
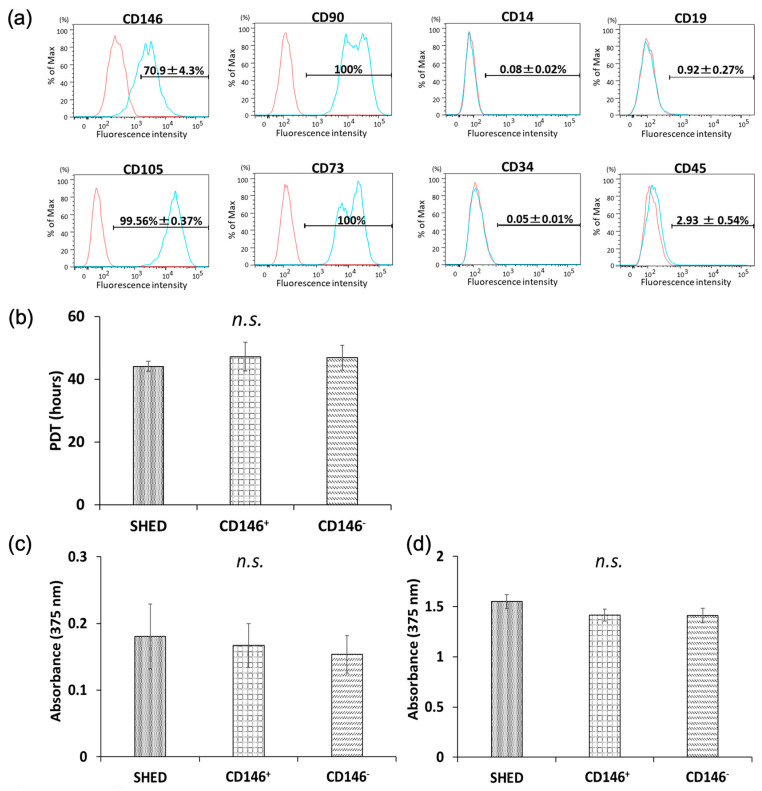
Surface-antigen analysis of SHED and comparative properties of SHED, CD146 + SHED, and CD146-SHED cellular proliferative potential. (**a**) Surface-antigen analysis of SHED. Among the heterogeneous population of SHED isolated from the deciduous pulp, 70.9 ± 4.3% of cells expressed CD146. MSCs were positive for CD90, CD73, and CD105 but negative for CD14, CD19, CD34, and CD45. (**b**) Comparison of population doubling time (PDT). The log phase was assessed based on the cell growth curve from day 2 to day 6 of incubation, and the PDT was calculated for this period. PDT did not differ significantly among the three groups (*n* = 5, Kruskal–Wallis test, Not significant; N.S.) (**c**,**d**). Comparison of cell proliferation. (**c**) Two hours after BrdU treatment, DNA synthesized in SHED was slightly higher than that in CD146 + SHED and CD146-SHED, but there were no significant differences among the three groups. (**d**) Twenty-four hours after BrdU treatment, the results were similar (*n* = 5, Kruskal–Wallis method, Not significant; N.S.).

**Figure 2 ijms-24-04048-f002:**
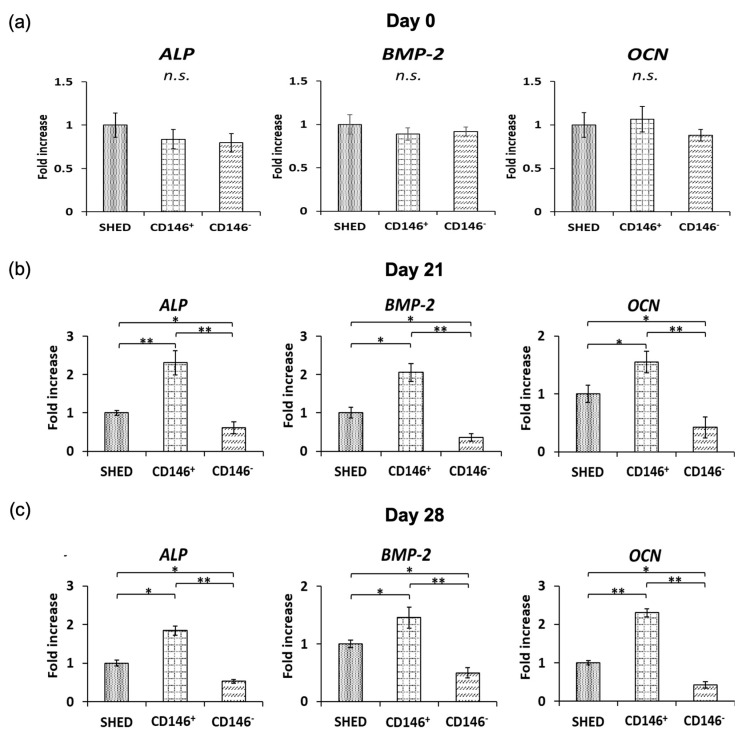
Gene expression in relation to bone differentiation on days 0, 21, and 28 of induction of bone differentiation. (**a**) Uninduced osteogenic differentiation. No significant differences were found among SHED, CD146 + SHED, and CD146-SHED in the gene expression of ALP, BMP-2, and OCN (*n* = 5, Not significant; N.S.). (**b**,**c**). Induced osteogenic differentiation on days 21 and 28. CD146 + SHED showed significantly higher gene expression of ALP, BMP-2, and OCN than SHED and CD146-SHED (*n* = 5, Kruskal–Wallis method, ** *p* < 0.01, * *p* < 0.05).

**Figure 3 ijms-24-04048-f003:**
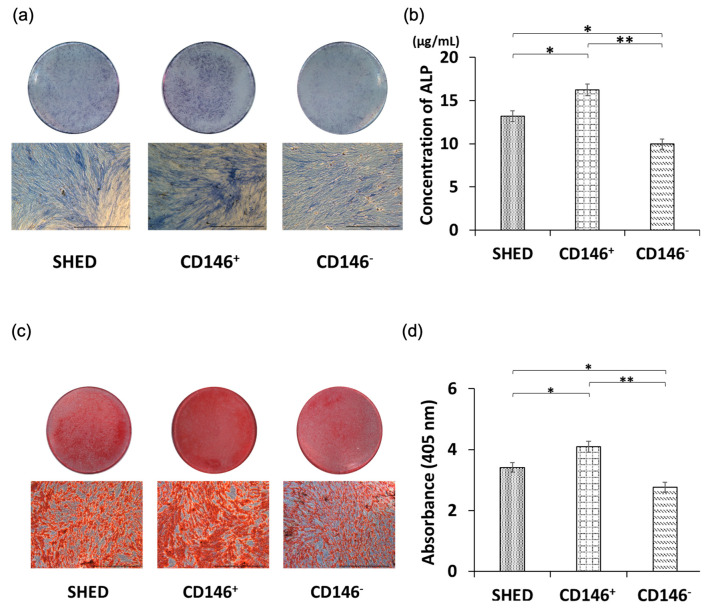
Osteogenic differentiation potential of SHED, CD146 + SHED, and CD146-SHED. (**a**) Osteogenic differentiation assessed by ALP staining on day 21. ALP staining was confirmed in all cells. The ALP staining intensity decreased in the following order: CD146 + SHED, SHED, and CD146-SHED (Scale bar = 500 μm). (**b**) Determination of ALP protein levels by ELISA. ALP protein levels were significantly higher in CD146 + SHED than those in SHED and CD146-SHED. The same was true for SHED compared to CD146-SHED (*n* = 5, Kruskal–Wallis method, ** *p* < 0.01, * *p* < 0.05). (**c**) Induced osteogenic differentiation. Osteogenic differentiation assessed by Alizarin red staining on day 28 decreased in the following order: CD146 + SHED, SHED, and CD146-SHED (Scale bar = 500 μm). (**d**) CD146 + SHED showed significantly higher absorbance values than SHED and CD146-SHED (*n* = 5, Kruskal–Wallis method, ** *p* < 0.01, * *p* < 0.05).

**Table 1 ijms-24-04048-t001:** The sequence of each primer.

Gene		Sequence (5′→3′)
GAPDH	Forward	CCA CTC CTC CAC CTT TGA
	Reverse	CAC CAC CCT GTT GCT GTA
ALP	Forward	ATG GTG GAC TGC TCA CAA C
	Reverse	GAC GTA GTT CTG CTC GTG GA
BMP-2	Forward	AAC ACT GTG CGC AGC TTC C
	Reverse	CTC CGG GTT GTT TTC CCA C
OCN	Forward	GCA GAG TCC AGG AAA GGG TG
	Reverse	GTC AGC AAC TCG TCA CAG

## Data Availability

The data supporting the findings of this study are available from the corresponding author upon reasonable request.

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
