# Peer review of "Bone Differentiation Ability of CD146-Positive Stem Cells from Human Exfoliated Deciduous Teeth"

_ijms, 2023, doi:10.3390/ijms24044048_

Round 1

Reviewer 1 Report

This manuscript should be revised with attention given to the overall paragraph structure and focus.  For example, in the first paragraph of the introduction, a description is given of tissue regeneration, including the need for scaffolds and function of the regenerated tissues;  yet this (scaffolds and function) were not the focus of the studies reported in this manuscript.  The paragraph goes on to introduce the use of cytokines and then MSC (3 different topics within one paragraph); all leaving the reader wondering what the focus is.  

The second paragraph states it is easier to harvest MSCs from exfoliated deciduous teeth,  and then goes on to focus on both SHED and DPSCs. The flow from one sentence to the other is not easy to follow.  It seems the authors are introducing the concept that SHED and DPSCs both include MSCs, but not much else is discussed as to their differences, other than SHED may be more convenient to utilize.  Are there any other reasons to focus on SHED?  Should similar studies be done with DPSC?  If so, this could be added into the discussion?

Please edit the introduction including the flow of the sentences and paragraphs so that the reader can easily follow both the background; what is unique in the studies reported in this manuscript; and the overall focus and rationale, which seems to be determining the outcome of adding CD146 to cultured SHED prior to transplant??

Materials and Methods:  

The methods for the Fluorescent cell sorting are confusing.  It seems that first the SHED cells were sorted for MSC positive, or negative, and both groups were further sorted for CD146 positive?  Which of these cell groups was then cultured with or without CD146 ?  Which cell group are the referred to as SHED (the unsorted cells?).  Please rewrite to make clear and describe the sequence of methods used and when sorting was done, and which cells were used for the subsequent analyses. 

Results:

In the methods MCSs were defined as being positive for CD90, CD75, and CS105.  According to Fig 1, then nearly 100% of SHED are MSC?  An important component of this appears to be that the cells were first grown on culture dishes before being sorted.  This should be clearly stated early in the methods , expanding on the description: “deciduous pulp extracted during orthodontic treatments were passaged and subjected to 112 flow cytometry.”  How were the cells cultured before passaging?

Discussion:

Paragraph 2 of the discussion appears to have many repeat sentences/phrases.  Please simplify; state the findings and how they related to previous findings in the literature?

For  the proliferation studies, are these results similar to other studies of stem cells supplemented with CD146?  It is important to put this result into context.

The paragraph beginning at line 390 focuses on VEGF and VEGFR, and yet VEGF was not analyzed in this study.  This is a very long paragraph that again is difficult to follow and doesn’t not seem to be based on the results of this study.  Therefore, it is not clear what this paragraph adds.  

The results of this study appear to be fairly simple, but yet important, including that all cultured SHED have MSC characteristics, and that addition of CD146 enhances osteogenic potential in vitro.  These results should be clearly stated and put into the context of previously published studies.

Author Response

Comments and Suggestions for Authors

This manuscript should be revised with attention given to the overall paragraph structure and focus. For example, in the first paragraph of the introduction, a description is given of tissue regeneration, including the need for scaffolds and function of the regenerated tissues; yet this (scaffolds and function) were not the focus of the studies reported in this manuscript. The paragraph goes on to introduce the use of cytokines and then MSC (3 different topics within one paragraph); all leaving the reader wondering what the focus is.

The second paragraph states it is easier to harvest MSCs from exfoliated deciduous teeth, and then goes on to focus on both SHED and DPSCs. The flow from one sentence to the other is not easy to follow. It seems the authors are introducing the concept that SHED and DPSCs both include MSCs, but not much else is discussed as to their differences, other than SHED may be more convenient to utilize. Are there any other reasons to focus on SHED? Should similar studies be done with DPSC? If so, this could be added into the discussion?

Please edit the introduction including the flow of the sentences and paragraphs so that the reader can easily follow both the background; what is unique in the studies reported in this manuscript; and the overall focus and rationale, which seems to be determining the outcome of adding CD146 to cultured SHED prior to transplant??

 [Response]: We sincerely thank you for providing critical comments and useful suggestions that have helped us considerably improve our manuscript. First, we deleted the corresponding text concerning thee scaffold content, tissue engineering, and cytokine from the Introduction section to prevent confusion as recommended by the reviewer. In the following paragraphs, we compiled the sentences holistically for better understanding of the background of DPSCs and SHED. Kindly check Page, 2 Line65-76. In addition, we have cited the article that compares SHED with DPSCs and described the background and benefits of SHED. The rationale for focusing on CD146 is described in Page 2, Line 79-95.

Materials and Methods:

The methods for the Fluorescent cell sorting are confusing. It seems that first the SHED cells were sorted for MSC positive, or negative, and both groups were further sorted for CD146 positive? Which of these cell groups was then cultured with or without CD146 ? Which cell group are the referred to as SHED (the unsorted cells?). Please rewrite to make clear and describe the sequence of methods used and when sorting was done, and which cells were used for the subsequent analyses. 

 [Response]: Thank you for your insightful suggestion. Each SHED collected from 5 patients was cultured and passaged to P3.Then, flow cytometry was performed using one of the 10 cm dishes in which the cells of each donor were cultured (five in total), and CD146, CD90, CD14, CD19, CD105, CD73, CD34, and CD45 were analyzed. The rest of the cells were not used for flow cytometry and were used for subsequent experiments as an unsorted SHED group. The representative results of one of the five donors are shown in Figure 1. Following the analysis of these surface antigens by flow cytometry, the cells were merely sorted for CD146 only into CD146-positive and CD146-negative SHEDs. Therefore, surface antigens other than CD146 were only analyzed and not sorted. In subsequent experiments, three groups of CD146-positive and CD146-negative SHEDs isolated by cell sorting and unsorted SHEDs without flow cytometry were used.

Kindly check Page3, Line100-101 and Page4 150-158.

Results:

In the methods MCSs were defined as being positive for CD90, CD75, and CS105. According to Fig 1, then nearly 100% of SHED are MSC? An important component of this appears to be that the cells were first grown on culture dishes before being sorted. This should be clearly stated early in the methods, expanding on the description: “deciduous pulp extracted during orthodontic treatments were passaged and subjected to 112 flow cytometry.” How were the cells cultured before passaging?

[Response]: Thank you for your pertinent question. According to the results in Figure 1, we assume that nearly 100% of SHED is MSC. SHED was isolated from the dental pulp in the same manner as by Miura et al. and Gronthos et al.

The following text elaborates on the isolation of SHED from the pulp. A mixture of α-MEM (Sigma-Aldrich, St. Louis, MO, USA), 4 mg/mL collagenase (Thermo Fisher Scientific, Waltham, MA, USA), and 3 mg/mL dispase (Godo Shusei, Tokyo) was prepared. The pulp tissue was immersed in the solution and minced with a scalpel. Adequate dental pulp tissue slices were transferred to a 10-ml tube and then incubated at 37° C under 5% CO2 for 20 minutes with shaking. Cell aggregates were eliminated using a 70-μm cell strainer (CORNING, Corning, NY, USA), and the filtered solution was diluted with α-MEM and centrifuged at 1,500 rpm for 5 minutes. The supernatant was aspirated and added to 20 % fetal bovine serum (FBS) (Daiichi Kagaku, Tokyo), 0.24 μl/ml kanamycin (Meiji Seika Pharma Co., Ltd., Tokyo), and 0.5 μl/ml penicillin (Meiji Seika Pharma). 1 μl/ml. After being suspended in α-MEM containing mL amphotericin (MP Biomedicals), they were seeded in a cell culture petri dish (CORNING) with a diameter of 35 mm, cultivated at 37°C and 5% CO2, and the cells were detached from the petri dish using PBS containing 0.25% trypsin (Nacalai Tesque, Kyoto) and 1 mM EDTA (Wako Pure Chemical Industries, Osaka) when confluent. and passaged. After the first passage (P1), the cells were cultured in α-MEM containing 10% FBS (Daiichi Kagaku, Tokyo) and the above antibiotics at 37°C under 5% CO2. The cell culture methods have been added to the Materials and methods section for better understanding of the reader. Kindly check Page 3, Line104-119.

Discussion:

Paragraph 2 of the discussion appears to have many repeat sentences/phrases. Please simplify; state the findings and how they related to previous findings in the literature?

 [Response]: We are very grateful for your suggestions. According to your instructions,

We deleted many repeating sentences/phrases in paragraph 2 of the discussion section; we have specifically deleted the following text:

In addition, bone regeneration and differentiation are promoted in CD146-positive BMSCs and umbilical cord-derived MSCs [32,33]. Moreover, in vivo validation analysis showed that the transplantation of CD146-positive SHED cells induced good bone regeneration, suggesting that CD146+SHED have higher bone regeneration ability and are suitable for transplantation into bone defects [34].

Kindly check.

For the proliferation studies, are these results similar to other studies of stem cells supplemented with CD146? It is important to put this result into context.

 [Response]: We are very grateful for your valuable suggestions. The following is a previous report of CD146 MSCs regarding as cell proliferation.

CD146+MSCs exhibited significantly higher cell proliferative capacity than CD146-MSCs in human endometrium-derived MSCs and periodontal ligament-derived MSCs [Fayazi M et al., Int J Reprod Biomed, 2016,14, 437-442; Zhu W et al., Arch Oral Biol, 2013, 58, 1791-1803].

However, some papers have demonstrated that CD146-MSCs have higher proliferative potential than CD146+MSCs [Espagnolle N et al., J Cell Mol Med, 2014, 18, 104-114; Paduno F et al.,Stem cell Rev, 2016, 12, 592-603]. Thus, the results vary based on the paper, which is also mentioned in [Paduno F et al.,Stem cell Rev, 2016, 12, 592-603].We could not find any studies that demonstrated no significant difference in the proliferative ability of CD146+MSCs and CD146-MSCs similar to our study. We have added the relevant information to the discussion section. Kindly see Page11, Line 389-396.

The paragraph beginning at line 390 focuses on VEGF and VEGFR, and yet VEGF was not analyzed in this study. This is a very long paragraph that again is difficult to follow and doesn’t not seem to be based on the results of this study. Therefore, it is not clear what this paragraph adds.

 [Response]: We greatly appreciate your valuable suggestion. As the reviewers have accurately pointed out, we did not examine the development of VEGF; thus, we deleted the discussion related to VEGF.

The results of this study appear to be fairly simple, but yet important, including that all cultured SHED have MSC characteristics, and that addition of CD146 enhances osteogenic potential in vitro. These results should be clearly stated and put into the context of previously published studies.

 [Response]: We are very grateful for your suggestions. We modified the content of the discussion items according to the reviewer’s suggestions. Kindly see Page12, Line414-438.

Other reviewers response.

We are very grateful to you for providing these useful comments, and appreciate your interest in our research. As indicated by the editor, we carefully modified the content of the entire article and discussion.

The manuscript has been rechecked for language and adherence to the formatting requirements.The manuscript has been proofread by a native English speaker (Editage Editing Services).

Kindly check our revised manuscript. We hope that our explanations and revisions are satisfactory. In addition, the manuscript has been proofread by a professional English language editing service provider. We have attached the certificate of proofreading for you to verify this claim.

We hope that the revised version of our paper is now suitable for publication in the International Journal of Molecular Sciences, and we look forward to hearing from you at your earliest convenience.

Reviewer 2 Report

This is a very well presented paper that attempts to look at the relevance of the CD146 (MCAM) surface marker on cells recovered from the dental pulp of deciduous teeth. There are some differences between CD146+ and CD146- cells from the dental pulp when cultured in osteogenic media.

The methods are straightforward, but there are a few problems. The number of donors are not given, although the number 5 is given for the number of donors of teeth for flow cytometry and sorting. Given the number of cells (1x106) in each of two solutions, I would suspect that the extracts from the five teeth were pooled. This should be stated explicitly if it is the case. 

The number of replicates used in the statistical analyses are not given. Also, if the assays were all performed on cultured pooled cells, whether sorted or not, then this needs to be explicitly stated as there is only one biological replicate.

In the results the numerical data are given to far too many significant figures. Although a common error, it should be corrected. With only 1 or a few biological replicates, means should only be reported to a maximum of two significant figures and standard deviation to one eg 14.28+/-0.745 micrograms per ml should be reported as 14 +/- 1.  There are 9 instances in the results section and three (without standard deviations) in the Discussion. Also, the population doubling times are all the same, there being no significant difference between 45, 47 and 47 hours respectively.

The first sentence of section 3.1 needs to be rewritten unless there were 10,000 donors used. The numbers reported come from FloJo's estimate of variance within a single run and are not valid standard deviations of a population of donors.

In the Discussion (top of page 11 of the proof, line 377) "detaining" should be "releasing".

Some 13 lines later (line 390) "... VEGF is not an abundant milieu;..." needs to be rewritten as VEGF is not a milieu. I presume that the authors wish to say that it is not as abundant in media as in vivo, however this would need a reference.

Six lines later (line 396 and 397) it is stated that "...the finding is consistent with that in this study, which showed that bone differentiation was enhanced in CD146+SHED with higher VEGF expression. VEGF was not assayed in this study and this conclusion cannot be made in the data presented.

Author Response

Comments and Suggestions for Authors

Reviewer 2#

This is a very well presented paper that attempts to look at the relevance of the CD146 (MCAM) surface marker on cells recovered from the dental pulp of deciduous teeth. There are some differences between CD146+ and CD146- cells from the dental pulp when cultured in osteogenic media.

 [Response]: Dear Reviewer #2, We thank you and the reviewers for the valuable comments and suggestions. Accordingly, the manuscript has been rechecked and the necessary changes have been made throughout the manuscript. Point-by-point responses to the comments have been prepared and attached herewith. The revisions made in response to the comments are marked using yellow highlights in the revised manuscript. We look forward to working with you to move this manuscript closer to publication in the International Journal of Molecular Sciences. Thank you for your consideration.

The methods are straightforward, but there are a few problems. The number of donors are not given, although the number 5 is given for the number of donors of teeth for flow cytometry and sorting. Given the number of cells (1x106) in each of two solutions, I would suspect that the extracts from the five teeth were pooled. This should be stated explicitly if it is the case. 

 [Response]: We are very grateful for your suggestions, which have helped us considerably improve our manuscript. SHEDs isolated from 5 patients were each cultured in 10-cm culture dishes until confluent. SHEDs derived from five donors were separately analyzed for surface antigens by flow cytometry. The flow cytometry results (Figure 1A) present one sample representative surface antigen rate among them. We analyzed SHED from 5 donors respectively by flow cytometry; however, we were unable to summarize these results in a histogram. Hence, we provide a representative sample of the flow cytometry results. Kindly check Page3 line100-101,Page 2 line 122-123, Page 2 Line 126-129, Page 4 line 150-158, Page 6 line 244.

The resulting SHED cells were cultured separately, and their ability to proliferate and differentiate into bone was examined for each of the five SHED cells of an individual. Therefore, n = 5. We revised the corresponding sentence to avoid misleading the reader. Kindly check Page3 line100-101,Page 2 line 122-123, Page 2 Line 126-129, Page 4 line 150-158, Page 6 line 244.

The number of replicates used in the statistical analyses are not given. Also, if the assays were all performed on cultured pooled cells, whether sorted or not, then this needs to be explicitly stated as there is only one biological replicate.

In the results the numerical data are given to far too many significant figures. Although a common error, it should be corrected. With only 1 or a few biological replicates, means should only be reported to a maximum of two significant figures and standard deviation to one eg 14.28+/-0.745 micrograms per ml should be reported as 14 +/- 1. There are 9 instances in the results section and three (without standard deviations) in the Discussion. Also, the population doubling times are all the same, there being no significant difference between 45, 47 and 47 hours respectively.

 [Response]: We thank you for providing critical comments and useful suggestions that have helped us considerably improve our manuscript. The resulting SHED cells were cultured separately. Thereafter, we examined the proliferative and osteogenic differentiation activities of each of the five each samples of SHED of an individual. Therefore, n = 5. Thus, the mean and standard deviation and the significance could be determined. We have added the relevant text to the Materials and Methods section to avoid misleading the readers. Kindly check Page3 line100-101,Page 2 line 122-123, Page 2 Line 126-129, Page 4 line 150-158

The first sentence of section 3.1 needs to be rewritten unless there were 10,000 donors used. The numbers reported come from FloJo's estimate of variance within a single run and are not valid standard deviations of a population of donors.

 [Response]: Thank you for your valuable suggestion.The number of donors is 1 only in Figure 1(a). Flow cytometry was performed in 5 donors; however, we were unable to express them collectively in a histogram. Hence, one representative donor is shown in Figure 1(a). For example, the figure for CD146 describes the percentage of positive cells as 70.9±4.3%. This indicates that the number of positive cells varies slightly with gating. Therefore, this ± notation does not indicate standard deviation. We have revised the text in the Materials and Methods section to avoid misleading the reader. Kindly check Page3 line100-101,Page 2 line 122-123, Page 2 Line 126-129, Page 4 line 150-158, Page 6 line 244.

In the Discussion (top of page 11 of the proof, line 377) "detaining" should be "releasing".

 [Response]: Thank you for your valuable suggestion. We have replaced “detaining” with “releasing” in the sentence accordingly. Kindly check Page12, Line399.

Some 13 lines later (line 390) "... VEGF is not an abundant milieu;..." needs to be rewritten as VEGF is not a milieu. I presume that the authors wish to say that it is not as abundant in media as in vivo, however this would need a reference.

 [Response]: We greatly appreciate your valuable suggestion. As the reviewers have pointed out, we did not examine the development of VEGF; thus, we omitted the discussion related to VEGF.

Six lines later (line 396 and 397) it is stated that "...the finding is consistent with that in this study, which showed that bone differentiation was enhanced in CD146+SHED with higher VEGF expression. VEGF was not assayed in this study and this conclusion cannot be made in the data presented.

 [Response]: We are very grateful for your valuable suggestions. As the reviewers have pointed out, we did not examine the development of VEGF; thus, we deleted the discussion related to VEGF.

Other reviewers response.

We are very grateful to you for providing these useful comments, and appreciate your interest in our research. As indicated by the editor, we carefully modified the content of the entire article and discussion.

The manuscript has been rechecked for language and adherence to the formatting requirements.The manuscript has been proofread by a native English speaker (Editage Editing Services).

Kindly check our revised manuscript. We hope that our explanations and revisions are satisfactory. In addition, the manuscript has been proofread by a professional English language editing service provider. We have attached the certificate of proofreading for you to verify this claim.

We hope that the revised version of our paper is now suitable for publication in the International Journal of Molecular Sciences, and we look forward to hearing from you at your earliest convenience.

Reviewer 3 Report

In the article: “Bone differentiation ability of CD146-positive stem cells from human exfoliated deciduous teeth”  the authors explored the effects of CD146 on cell proliferative and substrate metabolic abilities in a population of SHED.

Overall, this work is very interesting, the authors clearly explain the rational of the study and discussed the results. However, we would like to invite the authors  to clarify some points:

1.       Please check the language and punctuation;

2.       The reference 5, Langer and Vacanti is very old.  Is available a more recent to add?

3.       Among the introduction section, within the general description of pulp stem cells and their entailial to differentiate, the following reference should be useful; La Noce M, Stellavato A, Vassallo V, Cammarota M, Laino L, Desiderio V, Del Vecchio V, Nicoletti GF, Tirino V, Papaccio G, Schiraldi C, Ferraro GA. Hyaluronan-Based Gel Promotes Human Dental Pulp Stem Cells Bone Differentiation by Activating YAP/TAZ Pathway. Cells. 2021 Oct 26;10(11):2899. doi: 10.3390/cells10112899. PMID: 34831122; PMCID: PMC8616223;

4.       Why the cells were seeded on covered collagen I plate?

5.       The specific biomarkers of bone were checked also at protein level? For example by Western blotting?

6.       Osteopontin expression was not evaluated? Why?

7.       Figure 3 A and C: please specify the magnification used

Author Response

Comments and Suggestions for Authors

Reviewer 3#

In the article: “Bone differentiation ability of CD146-positive stem cells from human exfoliated deciduous teeth” the authors explored the effects of CD146 on cell proliferative and substrate metabolic abilities in a population of SHED.

Overall, this work is very interesting, the authors clearly explain the rational of the study and discussed the results. However, we would like to invite the authors to clarify some points:

 [Response]: Dear Reviewer #3, We thank you and the reviewers for the valuable comments and suggestions. Accordingly, the manuscript has been rechecked and the necessary changes have been made throughout the manuscript. Point-by-point responses to the comments have been prepared and attached herewith. The revisions made in response to the comments are marked using yellow highlights in the revised manuscript. We look forward to working with you to move this manuscript closer to publication in the International Journal of Molecular Sciences. Thank you for your consideration.

  1. Please check the language and punctuation.

 [Response]: Thank you for your insightful suggestion. We have revised the language and punctuation. Kindly check.

  1. The reference 5, Langer and Vacanti is very old. Is available a more recent to add?

 [Response]: Thank your valuable suggestion. Reviewer#1 pointed out that the Introduction text in addtion, the manuscript has been proofread by a native English speaker through Editage editing services. contained redundant information and included cytokine-based therapies and scaffolding and tissue engineering text, which may confound the reader. Thus, we substantially modified the introduction section and omitted the reference article from Langer et al. Kindly check page1,2 line 37-50.

  1. Among the introduction section, within the general description of pulp stem cells and their entailial to differentiate, the following reference should be useful; La Noce M, Stellavato A, Vassallo V, Cammarota M, Laino L, Desiderio V, Del Vecchio V, Nicoletti GF, Tirino V, Papaccio G, Schiraldi C, Ferraro GA. Hyaluronan-Based Gel Promotes Human Dental Pulp Stem Cells Bone Differentiation by Activating YAP/TAZ Pathway. Cells. 2021 Oct 26;10(11):2899. doi: 10.3390/cells10112899. PMID: 34831122; PMCID: PMC8616223.

 [Response]: We are very grateful for your suggestions. According to your instructions, the abovementioned reference has been added. Kindly check Page14, Line534-536.

  1. Why the cells were seeded on covered collagen I plate?

 [Response]: Thank you for your pertinent question. In our study group, cell cultures of MSCs cells were cultured in culture dishes coated with type 1 collagen since the engraftment rate of the cells is high; thus, stable cultures can be obtained.

  1. The specific biomarkers of bone were checked also at protein level? For example by Western blotting?

[Response]: I would like to thank you for this pertinent question. In our previous studies on the bone differentiation potential of MSCs, we were evaluating it using a multipurpose examination to comprehensively determine the bone differentiation potential (Huang YC et al., Cell Tissue Res 342;205-212:2010 : Kunimatsu R et al., Biochem Biophys Res Commun 18;193-198: 2018). With reference to previous reports, ALP quantitation and ALP stain were performed, and assessment of final osseous differentiation was performed using the ALZ stain. Protein expression could not be employed in this study since the funding of this fiscal year's grants had already been exhausted and ELISA and antibodies could not be purchased. Thank you for your understanding.

  1. Osteopontin expression was not evaluated? Why?

[Response]: Thank you for your insightful question. OPN and ALP are early osteogenes​is-related​ factors. OPN is involved in the remodeling seen in normal tissues, such as adhesion, migration, bone resorption, angiogenesis, and healing of wounds in various cells. It is also reportedly associated with bone destruction. OPN also plays crucial roles in the immune system, such as inducing TH1 immunity and activating macrophages, and it is considered to be a type of cytokine with various actions. Based on these backgrounds, we decided to evaluate ALP, BMP-2, and OCN purely to assess osteogenesis. Moreover, since ALP is a major early osteogenic marker, we confirmed early osteogenic markers in ALP but not in OPN in this study.

  1. Figure 3 A and C: please specify the magnification used

[Response]: We greatly appreciate your valuable suggestion. According to your instructions,

the magnification of Figure 3 A and C has been added. Kindly check Page10, Line327,332.

Other reviewers response.

We are very grateful to you for providing these useful comments, and appreciate your interest in our research. As indicated by the editor, we carefully modified the content of the entire article and discussion.

The manuscript has been rechecked for language and adherence to the formatting requirements.The manuscript has been proofread by a native English speaker (Editage Editing Services).

Kindly check our revised manuscript. We hope that our explanations and revisions are satisfactory. In addition, the manuscript has been proofread by a professional English language editing service provider. We have attached the certificate of proofreading for you to verify this claim.

We hope that the revised version of our paper is now suitable for publication in the International Journal of Molecular Sciences, and we look forward to hearing from you at your earliest convenience.